



**The impact of rheological uncertainty on dynamic topography**
**predictions: Gearing up for dynamic topography models**
**consistent with observations**
**Ömer F. Bodur, Patrice F. Rey**
EarthByte Group, School of Geosciences, The University of Sydney, NSW 2006, Australia,
*Correspondence to:* Ömer F. Bodur  (omer.bodur@sydney.edu.au)
**Abstract**
Much effort has been given on extracting the dynamic component of the Earth's topography,
which is driven by density heterogeneities in the mantle. Seismically mapped density
anomalies have been used as an input into mantle convection models to predict the present-
day mantle flow and stresses applied on the Earth's surface, resulting in dynamic topography.
However, mantle convection models give dynamic topographies generally larger by a factor
of ~2 compared to dynamic topographies estimated from residual topography after extraction
of the isostatically compensated topography. Our 3D thermo-mechanical numerical
experiments suggest that this discrepancy can be explained by the use of a viscosity model,
which doesn't account for non-linear viscosity behaviour. In this paper, we numerically
model the dynamic topography induced by a spherical density anomaly embedded into the
mantle. When we use non-linear viscosities, our numerical models predict dynamic
topographies lesser by a factor of ~2 than those derived from numerical models using
isoviscous rheology. This reduction in dynamic topography is explained by either the
formation of a low viscosity channel beneath the lithosphere, or a decrease in thickness of the
mechanical lithosphere due to induced local reduction in viscosity. Furthermore, we show
that uncertainties related to activation volume and fluid activity lead to variations in dynamic
topography of about 20%.



## 1. Introduction

The Earth's mantle is continuously stirred by hot upwellings from the core-mantle
boundary, and by subduction of colder plates from surface into the deep mantle (Pekeris,
1935; Isacks et al., 1968; Molnar and Tapponnier, 1975; Stern, 2002). This introduces
temperature and density anomalies that stimulate mantle flow and forces dynamic uplift or
subsidence at the plates' surface (Gurnis et al., 2000; Braun, 2010; Moucha and Forte, 2011;
Flament et al., 2013). Dynamic topography can affect the entire planet's surface with varying
magnitudes. Because it is typically a low-amplitude and long-wavelength transient signal, it
is often dwarfed by topography created by plate tectonic processes. Therefore, investigations
on dynamic topography signals mostly focus on non-tectonic regions where the dynamic
topography can be extracted from the subsidence history of sedimentary basins. Dynamic
subsidence and uplift events are identified by isolating part of the subsidence that cannot be
explained either by thermal relaxation or tectonic processes such as crustal thinning (Sclater
and Christie, 1980).

For the present day, the observational constraints on dynamic topography come from residual
topography measurements (Hoggard et al., 2016). Residual topography is calculated by
removing the isostatic components from the Earth's topography (Crough, 1983; Cazenave et
al., 1989; Davies and Pribac, 1993; Steinberger, 2007). Hoggard et al., (2016)'s
comprehensive work revealed that residual topography varies between ±500 m at very long-
wavelengths (i.e. ~10,000 km) and can increase up to ±1,000 m at shorter wavelengths (i.e.
~1,000 km). However, the accuracy of these estimates depends on our knowledge of the
thermal and mechanical structure of the lithosphere. Another approach to constrain present
day Earth's dynamic topography involves numerical modelling of present-day mantle flow
using seismically mapped density anomalies as an input (Steinberger, 2007; Moucha et al.,



2008; Conrad and Husson, 2009). However, this method requires a good knowledge of the
viscosity structure in the Earth's interior (Parsons and Daly, 1983; Hager, 1984; Hager et al.,
1985; Hager and Clayton, 1989). The problem is that dynamic topography predictions
derived from mantle convection models are generally larger by a factor of two than estimates
from residual topography (Cowie and Kusznir, 2018; Flament et al., 2013). We hypothesise
that this could be related to an oversimplification of the viscous behaviour of the flowing
mantle. In this paper, we explore how the magnitude of dynamic topography is impacted
when we use a viscosity model in which the viscosity depends on strain rate, temperature,
pressure and fluid content. In this paper, we first  summarize the well-established analytical
solution for calculating dynamic topography induced by a spherical density anomaly
embedded into an isoviscous fluid (Morgan, 1965a; Molnar et al., 2015). Then, assuming
isoviscous rheology, we illustrate that the amplitude of dynamic topography depends on the
viscosity structure of the Earth's interior as shown by (Morgan, 1965a; Molnar et al., 2015).
Finally, we use 3-D coupled thermo-mechanical numerical experiments of the Stokes' flow to
assess the dependency of dynamic topography to nonlinear rheology using viscosity which
depends on temperature, pressure, strain rate and fluid content. We show that more realistic
rheology can induce local variations in viscosity and result in lesser magnitude of dynamic
topography than those derived from models using isoviscous rheology.

**2. Dynamic topography driven by a rising sphere: Analytical and**
**numerical solutions**
**2.1 Analytical solution for one layer isoviscous fluid**

73         We assume here a simple 2D model representing a very viscous spherical density

anomaly embedded into a semi-infinite isoviscous fluid bounded by an upper free surface.
Earliest analytical investigations revealed that, albeit counter-intuitive, the magnitude of the



induced surface deflection due to the rising sphere is independent of the viscosity of the fluid.
The dynamic topography is a function of the vertical total stress ($\sigma_{zz}$) applied to the surface
which is proportional to the size and depth of the density anomaly according to Equation 1
(Morgan, 1965a, 1965b).
$\sigma_{zz}(x, 0) = [2g\delta\rho r^3]\dfrac{D^3}{(D^2+x^2)^{5/2}}$ (1),
where $g$ is the gravitational acceleration, $\delta\rho$ is density difference between the anomaly and
the ambient material, $r$ is radius of the sphere, and $D$ is distance from the surface to the centre
of the anomaly (modified from Morgan 1965a, see Figure 1a). The dynamic topography $e_{zz}$ is
given by:
$e_{zz}(x, 0) = \dfrac{\sigma_{zz\,(x,0)}}{\rho g}$  at $z = 0$ (2),
where $\rho$ is mantle density (Morgan, 1965a; Houseman and Hegarty, 1987). In Figure 1a, we
plot the dynamic topography induced by a sphere of 1% density anomaly, whose centre is at
372 km depth ($D$= 372 km) below the free surface. We calculate the vertical total stress and
convert it to dynamic topography by using Equation 2 for different values of the radius of the
sphere. The amplitude of dynamic topography shows an accelerating increase by cubic
dependence on the radius of the spherical density anomaly (Fig. 1a, black solid line). For the
same problem, Molnar et al., (2015) provided a solution allowing to consider density
anomalies of finite viscosity ($\eta_{sphere}$) (Eqn. 3):
$\sigma_{zz}(C, 0) = \dfrac{-\delta\rho r^3 D}{3f}\left[\dfrac{3-2f}{C^3} + \dfrac{18(f-1)r^2}{C^5} + \dfrac{6fD^2}{C^5} - \dfrac{30(f-1)r^2D^2}{C^7}\right]$ (3),
where $C^2 = D^2 + x^2$ and f = $(\eta_1 + \dfrac{3\eta_{sphere}}{2})/(\eta_1 + \eta_{sphere})$, for very viscous sphere ($\eta_{sphere} \gg$
$\eta_1$) $f$=1.5, and deformable sphere ($\eta_{sphere} \cong \eta_1$) $f$<1.5. In Figure 1a, we give two more plots
of dynamic topography where $f$= 1.5 for hard sphere and $f$=1.25 for $\eta_{sphere} = \eta_1$ by using
Equation 2 and 3. Figure 1a shows that a rising deformable sphere creates higher dynamic
topography compared to a very viscous sphere. These show that the viscosity contrast





between the spherical anomaly and the surrounding material can affect the dynamic
topography. In the section that follows, we explore how dynamic topography varies when
there is layering in viscosity, such as existence of a strong lithosphere above the anomaly.

**2.2 The impact of layered viscosity structure on dynamic topography**
A more generalized solution has been put forward to accommodate the presence of a
stronger upper layer representing the lithosphere ($\eta_2$) above a lower weaker layer
representing the convective mantle ($\eta_1$, with $\eta_1<\eta_2$ in Fig. 1b). In this case, Morgan (1965a)
showed (Eqn. 4) that the normal total stress induced by the density anomaly at depth is
dependent on the mass anomaly per unit length ($M_u$), its depth ($D$), and marginally on the
ratio of the viscosity of the convective mantle to the viscosity of the lithosphere ($R=\eta_1/\eta_2$).
$\sigma_{zz}(x,0) = \int_0^\infty \sigma_n \cos nx \ dn$  (4),
where
$\sigma_n = \dfrac{M_u g e^{-n(D-d)}}{2\pi(RS_h + C_h)}\left\{1 + n(D-d) + nd\left[\dfrac{1 - nD + n(D-d)(RC_h + S_h)/(RS_h + C_h)}{1 + nd(1 - R^2)/(RS_h + C_h)(RS_h + S_h)}\right]\right\}$
and $C_h=cosh \ nd$, and $S_h=sinh \ nd$ ($n$ is wave number) and $d$ is the upper layer thickness
(modified from Morgan 1965a). Following Morgan (1965a), Figure 1b illustrates the relative
importance of $R$ as well as the ratio of the thickness of the upper layer to the depth of the
anomaly ($d/D$). As long as the lithosphere is more viscous than the asthenosphere, the vertical
total stress at the surface has a minor dependence on the viscosity of the lithosphere (see solid
lines with $R=1$ and $R=0.01$ in Fig. 1b). Figure 1b also shows that the magnitude of dynamic
topography increases as the density anomaly is brought closer to the surface (compare solid
black line with R=1 and dashed black line with R=1). Moreover, its sensitivity on the relative
viscosity of the lithosphere also increases. Under the assumption where the lithosphere is less
viscous than the asthenosphere, the normal stress is much reduced and is strongly dependent



on the viscosity of the lithosphere (Fig. 1b). These demonstrate that layering in viscosity can
have strong impact on the amplitude of dynamic topography. In the next section, we use
analytical solutions above to benchmark a numerical model, which we will then extend to
non-linear viscosity.

**2.3 Numerical solutions**
For comparison with analytical solutions (Morgan, 1965a; Molnar et al., 2015), we
consider 3D numerical models involving 1, 2 and 3 isoviscous layers. These benchmark
experiments will be used as references for the non-isoviscous models in section 3. Taking
advantage of the symmetry of the experimental setup, we extract viscosity and velocity fields
along a 2D cross section passing through the centre of the thermal anomaly, from which we
get streamlines and vertical velocity profile along the vertical axis at the centre of the models.
We calculate the dynamic topography from the normal stress computed at the surface. We
use the open-source code Underworld which solves the Stokes' equation at insignificant
Reynolds value (Moresi et al., 2003, 2007). The 3D computational grid represents a domain
3,840 km x 3,840 km x 576 km with a resolution of 6 km along the vertical $z$ axis and 10 km
along the $x$ and $y$ axes (Fig. 2). In all experiments, we include a 42 km thick continental crust
above the upper mantle. The density structure (see Table 1 for all parameters thermal
parameters) is sensitive to the geotherm via a coefficient of thermal expansion and
compressibility. The geotherm is defined using a radiogenic heat production in the crust, a
constant temperature of 20ºC at the surface, and a constant temperature of 1,350ºC at 150 km.
We disregard the adiabatic heating and the asthenosphere is kept at 1,350ºC. At a depth of
372 km below the surface, we embed a positive spherical temperature anomaly of +324ºC
which delivers a 1% volumetric density difference. The radius of the sphere is 96 km. In all



experiments, we impose free slip velocity boundary conditions at all walls, such as $V_x$ and $V_y$
are set to be free, but $V_z$=0 cm yr$^{-1}$ at the top wall.

### 2.3.1 Dynamic topography due to a rising sphere in an isoviscous fluid

In the first experiment (Fig. 3a Experiment 1), we assign the same constant depth-
independent viscosity of $10^{21}$ Pa s to the crust, mantle and the density anomaly. The
streamlines for Experiment 1 (Fig. 3a) show formation of two convective cells at the sides of
the sphere covering the entire crust and mantle. The vertical velocity profile indicates that the
thermal anomaly is rising with a peak velocity of ~2.4 cm yr$^{-1}$, which is faster than the 2.0 cm
yr$^{-1}$ predicted by the analytical solution (Fig. 4a). Experiment 1 predicts a dynamic
topography of 114 m (Fig 4b) which is lower than 132 m predicted by Molnar et al., (2015)'s
analytical solution. We have verified that increasing the depth of our model from 576 km to
864 km increases the dynamic topography from 114 m to 122 metres. Therefore, the misfit in
amplitude of dynamic topography arises from the finite space in our numerical experiments
compared to the semi-infinite half-space used in the analytical solution. Our numerical
experiment using isoviscous material delivers a result globally consistent with the analytical
solutions of Morgan (1965a) and Molnar et al., (2015).

### 2.3.2 Dynamic topography on a strong lithosphere above an isoviscous asthenosphere

In Experiment 2, we assign to the lithosphere a constant viscosity 100 times larger
(i.e. $10^{23}$ Pa s) than that of the asthenosphere (i.e. $10^{21}$ Pa s, Fig. 3b). This layering results in a



decrease in thickness of the asthenosphere. As a result, the convective cells are narrower (Fig.
3b). The streamlines are deflected across the lithosphere-asthenosphere boundary due to
viscosity contrast (Fig. 3b), and there is a sharp variation in vertical velocity at the base of the
lithosphere (Fig. 4a, red solid line). The maximum vertical velocity of ~2.1 cm yr$^{-1}$ is attained
near the centre of the anomaly. When compared to Experiment 1, the dynamic topography
(Fig. 4b, red solid line) shows a significant increase from ~114 m to ~174 m. This increase is
consistent with analytical estimations showing an increase in dynamic topography for the
case where viscosity increases toward the surface (see Fig. 1b, R<1). In Experiment 2a (not
shown here), we tested a different ratio of thickness of the lithosphere to the depth of the
anomaly (see *d/D* in Equation 4) by increasing the lithospheric thickness from 150 km to 200
km, while keeping all parameters identical to those of Experiment 2. As predicted by Eqn. 4,
the model gives a dynamic topography of ~191 m, the highest among all experiments (Fig.
4b, red dashed line). Overall, counter-intuitively, the presence of a thick viscous lithosphere
enhances the dynamic topography.

### 185   2.3.3 The impact of low viscosity channel on the dynamic topography

186       In Experiment 3 (Fig. 3c), we introduce a third 60 km thick low viscosity layer (i.e.

10$^{19}$ Pa s) beneath the base of the lithosphere. The existence of a low viscosity layer has been
suggested by several works (Craig and McKenzie, 1986; Phipps Morgan et al., 1995;
Stixrude and Lithgow-Bertelloni, 2005; Becker, 2017). In this experiment, in order to prevent
large viscosity contrast that can impede the numerical convergence, the viscosity of the
lithosphere and ambient asthenosphere are set as 10$^{22}$ Pa s and 10$^{21}$ Pa s, respectively. When
compared to Experiment 1, streamlines indicate a further decrease in size of the convective
cells, and more importantly, strong horizontal divergence of the streamlines within the low



viscosity layer (Fig. 3c). The vertical velocities are also enhanced in the asthenosphere, and
reach up to ~2.8 cm yr$^{-1}$ slightly above the centre of the anomaly (Fig. 4a, orange solid line).
When compared to Experiment 1, we observe a strong reduction in dynamic topography (Fig.
4b, orange solid line) from 114 m to 88 m. This is due to the damping effect of the low
viscosity channel, which reduces the deviatoric stress through its ability to flow. This low
viscosity channel acts as a decoupling layer.
Until now, the viscosities were assumed to be constant. However, results from experimental
deformation on mantle aggregates strongly suggest that the viscosity is highly nonlinear
(Hirth and Kohlstedt, 2003). In what follows, we explore the influence of more realistic
viscosities on dynamic topography.

**3. The impact of nonlinear viscosity on dynamic topography**
**3.1 Viscosity structure of the Earth's interior**
Earth's mantle is not isoviscous. Geological records of relative sea level changes
related to postglacial rebound, geophysical observations of density anomalies inferred from
seismic velocity variations in the mantle and satellite measurements of the longest
wavelength components of the Earth's geoid have been used to infer the radial viscosity
profile of the Earth's interior (Hager et al., 1985; Forte and Mitrovica, 1996; Mitrovica and
Forte, 1997; Kaufmann and Lambeck, 2000). Henceforward, beneath the lithosphere, a
variation in viscosity up to two orders of magnitude has been proposed (e.g., Kaufmann and
Lambeck, 2000). Investigations of the rheological properties of crustal and mantle rocks via
rock deformation experiments revealed a nonlinear dependence of viscosity on applied
deviatoric stress, pressure, temperature, grain size and the presence of fluids (Post and
Griggs, 1973; Chopra and Paterson, 1984; Karato, 1992; Karato and Wu, 1993; Gleason and



Tullis, 1995; Ranalli, 1995; Hirth and Kohlstedt, 2003; Korenaga and Karato, 2008). These
experiments lead to the following relationship:
$\eta_{eff}(\dot{\varepsilon}, P, T) = A^{\left(\frac{-1}{n}\right)} d^{\left(\frac{p}{n}\right)} f_{H_2O}^{\left(\frac{-r}{n}\right)} \dot{\varepsilon}^{\left(\frac{1}{n}-1\right)} e^{\left(\frac{Q+PV}{nRT}\right)}$ (5).
where σ, ε̇ and $A$ stands for differential stress, strain rate and pre-exponential factor; $p$, $r$ and
$n$ are exponents for grain size ($d$), water fugacity ($f_{H_2O}$) and stress, respectively; $V$ and $Q$ are
the energy and volume of activation.

In the case where mantle flow is driven by the temperature difference at the boundary of the
convective layer or by internal heating, the dominant strain mechanism is diffusion creep
because low deviatoric stresses are expected. However, mantle flow in the vicinity of a
moving density anomaly is likely driven by deviatoric stresses that exceed the threshold for
dislocation creep. In this case, nonlinear viscosities lead to strong local variation in viscosity
in the vicinity of the moving density anomaly. Are those local variations in viscosity
important for dynamic topography? To answer this question, we need reasonable constraints
on the rheological parameters controlling rocks' viscosity. However, the extrapolation from
laboratory strain rates typically in the range of $10^{-6}$ s$^{-1}$ to $10^{-4}$ s$^{-1}$ to mantle conditions where
strain rates are typically on the order of $10^{-13}$ s$^{-1}$ results in significant uncertainties on the
activation volume, activation energy and stress exponent (Hirth and Kohlstedt, 2003;
Korenaga and Karato, 2008). In what follows, we explore how nonlinear viscosity impacts
the dynamic topography and address how the uncertainties on the activation volume can
affect the dynamic topography.

In Experiments 4 and 5 (Fig. 5), we use published visco-plastic rheological parameters for the
crust and mantle, therefore the viscosity depends on temperature, pressure and strain rate as



indicated by Equation 5. We use quartzite rheology for the crust (Ranalli, 1995), and test both
dry and wet olivine rheologies for the upper mantle (Hirth and Kohlstedt, 2003). Other
parameters are identical to those in Experiments 1-3. We give all the rheological and thermal
parameters in Table 1. For a particular rheology (i.e. dry or wet) we vary the activation
volume by using the minimum and maximum reported values (Hirth and Kohlstedt, 2003).

**3.2 Numerical results: the case of dry olivine**
In Experiments 4a and 4b, we consider dry dislocation creep of olivine ($n > 1$, $p = 0$, r
$= 0$) in the mantle. The reported activation volume for this rheology varies between $6 \times 10^{-6}$ m$^3$
mol$^{-1}$ and $27 \times 10^{-6}$ m$^3$ mol$^{-1}$ (Hirth and Kohlstedt, 2003). In Experiment 4a (Fig. 4b), we test
the lower value. The streamlines show similar pattern with Experiment 2. Interestingly, the
maximum vertical velocity peaks at 75 cm yr$^{-1}$, near the upper boundary of the sphere (Fig.
6a, black dashed line). This is due to the formation of a low viscosity asthenosphere above
the rising sphere (Fig. 5a, Experiment 4a). This experiment gives a dynamic topography of
~149 m (Fig. 6b, black dashed line). It confirms that a strong contrast in viscosity between
the lithosphere and asthenosphere enhances the dynamic topography signal. We note that the
viscosity contrast is attained by smoother transition between the lithosphere and
asthenosphere (Fig. 7a, black dashed line). This also effectively reduces the thickness of the
lithosphere below 140 km, which is 150 km thick by the thermal definition (Fig. 7c).

When we increase the activation volume to $27 \times 10^{-6}$ m$^3$ mol$^{-1}$, the convection cells grow much
larger and show continuity through the lithosphere (Fig. 5a, Experiment 4b). The sphere has a
very low rising speed of ~0.25 cm yr$^{-1}$ (Fig. 6a, black solid line). Compared to Experiment
4a, the dynamic topography shows a strong decrease from ~149 m to ~105 m (Fig. 6b, black



solid line). This is an example where the system behaves nearly as a single layer with
homogenous viscosity. The near absence of viscosity contrast between the lithosphere and
asthenosphere explains the smaller magnitude of the dynamic topography. Moreover, the
formation of moderately low viscosity channel (Fig. 7a, black solid line) also contributes to
the decrease of the dynamic topography.

**3.3 Numerical results: the case of wet olivine**
In Experiments 5a and 5b, we consider dislocation creep of wet dry olivine in the
mantle. The reported uncertainty in activation volume is between $11 \times 10^{-6}$ $m^3$ $mol^{-1}$ and
$33 \times 10^{-6}$ $m^3$ $mol^{-1}$ (Hirth and Kohlstedt, 2003). In Experiment 5a, we test the lower value. The
streamlines show similar pattern with Experiment 4a, but with slightly larger convective cells
(Fig. 5b, Experiment 5a). The rising speed of the anomaly exceed 140 cm $yr^{-1}$ (Fig. 6a,
orange dashed line). This is promoted by the low viscosity region sitting above the rising
sphere. The dynamic topography is ~110 m (Fig. 6b, orange dashed line). This is a bit
surprising because of the strong (3 orders of magnitude) contrast in viscosity between the
lithosphere and asthenosphere. However, Figure 7a shows that thickness of the viscous
lithosphere is reduced by about 30 to 45 km in comparison to Experiment 4a (10 – 30 km)
which delivered a dynamic topography of ~149 m with same viscosity contrast (Figure 7b,c).

In Experiment 5b, we increase the activation volume from $11 \times 10^{-6}$ $m^3$ $mol^{-1}$ to $33 \times 10^{-6}$ $m^3$
$mol^{-1}$. The vertical velocities show significant decrease from 140 cm $yr^{-1}$ to 0.34 cm $yr^{-1}$ (Fig.
6a, orange solid line). This is due to an increase in viscosities above the rising sphere.
Compared to Experiment 5a, the dynamic topography decreases from ~110 m to ~90 m (Fig.
6b, orange solid line). Compared to Experiment 4b, the dynamic topography is expected to be
higher due to slight increase in viscosity contrast (Fig. 7a,b). However, the increase in



thickness of the low viscosity channel (Fig. 7a,d) is more effective and thereby causes a
greater reduction in magnitude of the dynamic topography.

In summary, experiments using nonlinear rheology generally give lower amplitudes of
dynamic topography compared to experiments using isoviscous rheology (Fig. 8). When we
use dry olivine rheology for the upper mantle, the dynamic topography varies between ~105
m and ~149 m, whereas under wet conditions, the dynamic topography varies between ~90 m
and ~110 m (Fig.8). These variations are due to uncertainties in the activation volume as well
as fluid content in olivine rheologies.

## 301    4. Discussion and conclusion

302       By using coupled 3D thermo-mechanical numerical experiments, we model the

dynamic topography driven by a rising sphere of 1% density anomaly, having 96 km radius
and emplaced at 372 km depth. In line with analytical studies (Morgan, 1965a; Molnar et al.,
2015), the experiments show that dynamic topography is sensitive to viscosity contrast
between the lithosphere and asthenospheric mantle above the rising anomaly, and the
thickness of the lithosphere (Fig. 7). Higher viscosity contrasts result in amplification of the
dynamic topography (Fig. 7a,b), whereas formation of a low viscosity channel reduces the
dynamic topography (Fig. 7a,d). The experiments using nonlinear rheologies show local
variations in viscosity, which contribute to the dynamic thinning of the mechanical
lithosphere and causes reduction in dynamic topography. In addition, models using high-
activation volume creates low viscosity channel above the density anomaly, which
contributes decreasing the dynamic topography.



Predictions of dynamic topography derived from mantle convection models are compared
against residual topography which is the component of Earth's topography that is not
compensated by crustal isostasy (Flament et al., 2013; Hoggard et al., 2016). In a recent work
(Cowie and Kusznir, 2018), it has been argued that dynamic topography predictions require
scaling of amplitudes by ~0.75 to match with residual topography (Flament et al., 2013;
Steinberger et al., 2017; Cowie and Kusznir, 2018). When density anomalies are shallower
than 220 km, the misfit increases demanding a scaling factor of ~0.35 (Steinberger et al.,
2017; Cowie and Kusznir, 2018). Our numerical experiments show that amplitude of
dynamic topography can be nearly halved (e.g. from ~174 m in Exp. 2 to ~90 m in Exp. 5b)
when we consider non-linear rheology. Therefore, we propose that part of the misfit between
the dynamic topography extracted from numerical modelling of mantle convection and
dynamic topography estimated from residual topography can be explained by the
oversimplification of mantle viscosity in convection models. Moreover, if the density sources
are shallower, the dynamic topography becomes more sensitive to the viscosity structure as
has already been shown by (Morgan, 1965a; Hager and Clayton, 1989) and may lead to
higher misfits.

As shown in Figure 8, uncertainties on the activation volume result in variation in dynamic
topography which are higher in experiments using dry olivine rheology (i.e. 17% from an
average of ~127 m) compared to experiments using wet olivine rheology (10% from an
average of ~100 m). The comparison between numerical experiments using dry olivine (Exp.
4a) and wet olivine (Exp. 5b) indicates that the variation in dynamic topography can be as
much as 25% from an average of ~120 m. These variations can be lessened if we have better
constraints on the mantle rheology, which will advance the dynamic topography models as
well as our understanding of the interaction between deep mantle and the Earth's surface.

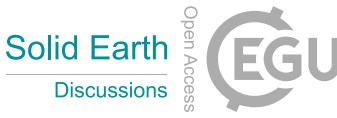

**Figures and Captions**

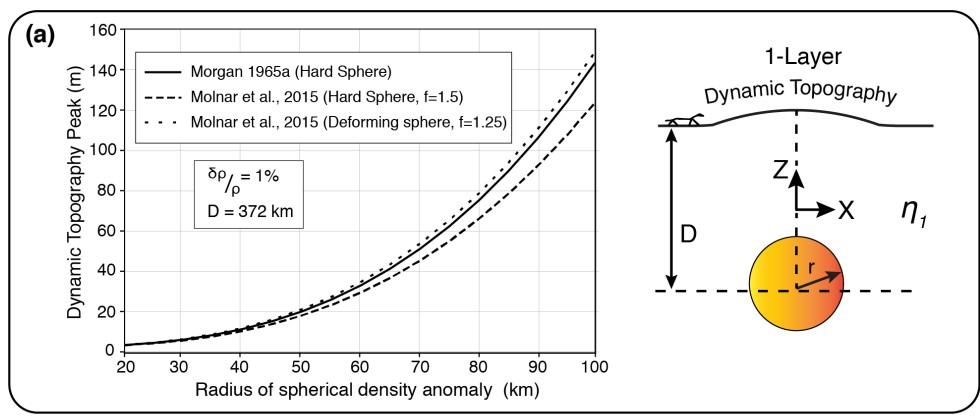

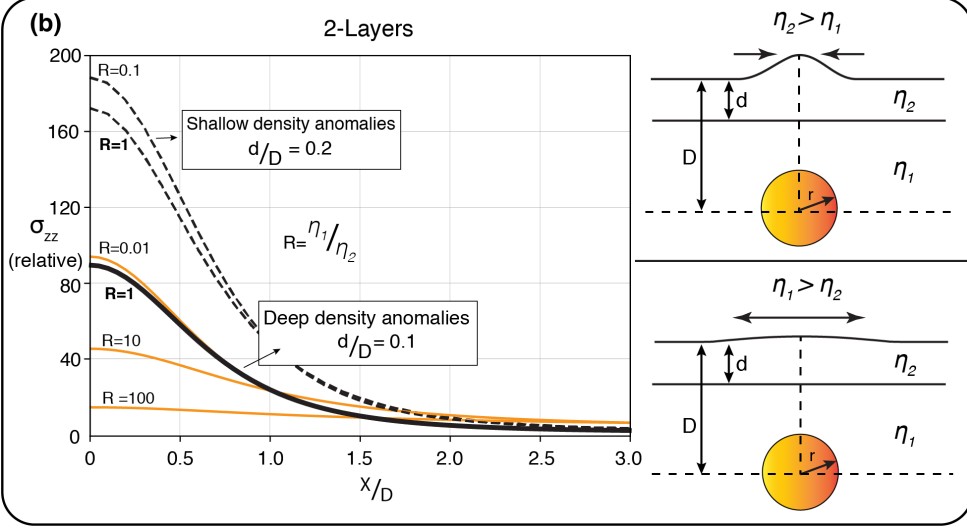


**Figure 1. Dynamic topography driven by a spherical density anomaly of radius *r* at**
**depth *D* embedded in a fluid whose viscosity structure is varied. (a) Variation in**
**dynamic topography by radius of a spherical 1% density anomaly centred at 372 km**
**depth in a single isoviscous fluid whose viscosity is $\eta_l$. The normal total stresses are**
**calculated by Equation 1 in the text taken from Morgan (1965a) (hard sphere), and**
**Equation 3 in the text taken from Molnar et al (2015) (hard and deforming spheres),**
**and converted to dynamic topography by using Equation 2. (b) The case where the fluid**





**is no longer a single layer, but is composed of two layers with viscosities $\eta_1$ and $\eta_2$ for**
**the lower and upper layers, respectively. We plot the variation in relative normal total**
**stress at the surface in half-space due to a spherical density anomaly at a depth *D* with**
**radius *r* by using Equation 4 in the text, taken from Morgan (1965a). The plots show**
**relative variation in stress at a relative distance of *X/D*, for different viscosity ratio of**
**the layers (*R=$\eta_1/\eta_2$*), as well as ratio of upper layer thickness to depth to the centre of**
**the anomaly (*d/D*) of which higher values correspond to more shallow density anomalies**
**or thicker lithosphere for constant depth (D).**

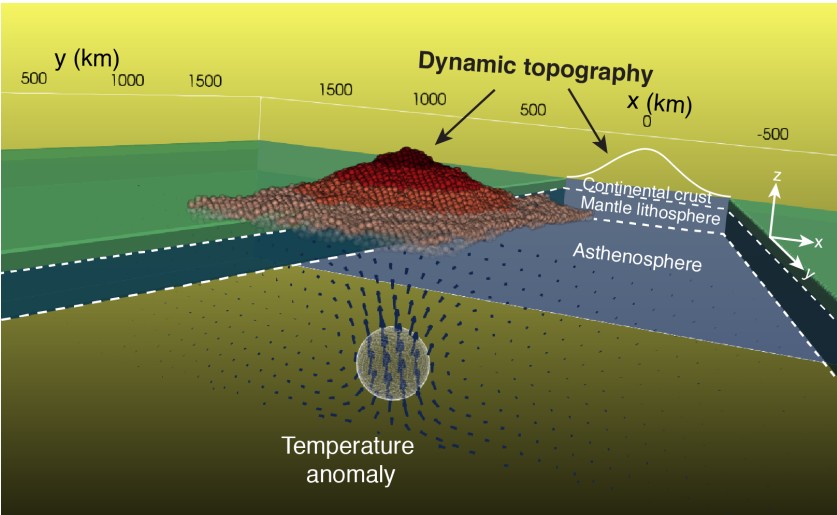


**Figure 2. 3D Numerical model of a spherical temperature anomaly having 96 km radius**
**and a density of 1% less dense than the ambient mantle embedded in a depth of 372 km.**
**The model space is 3,840 km long in *x* and *y* axes, and 576 km deep along the *z* axis. The**
**dynamic topography is depicted as an exaggerated surface on the top of the model and**
**is also reflected on the *x-z* plane.**

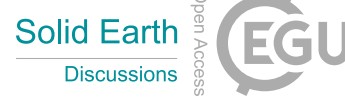

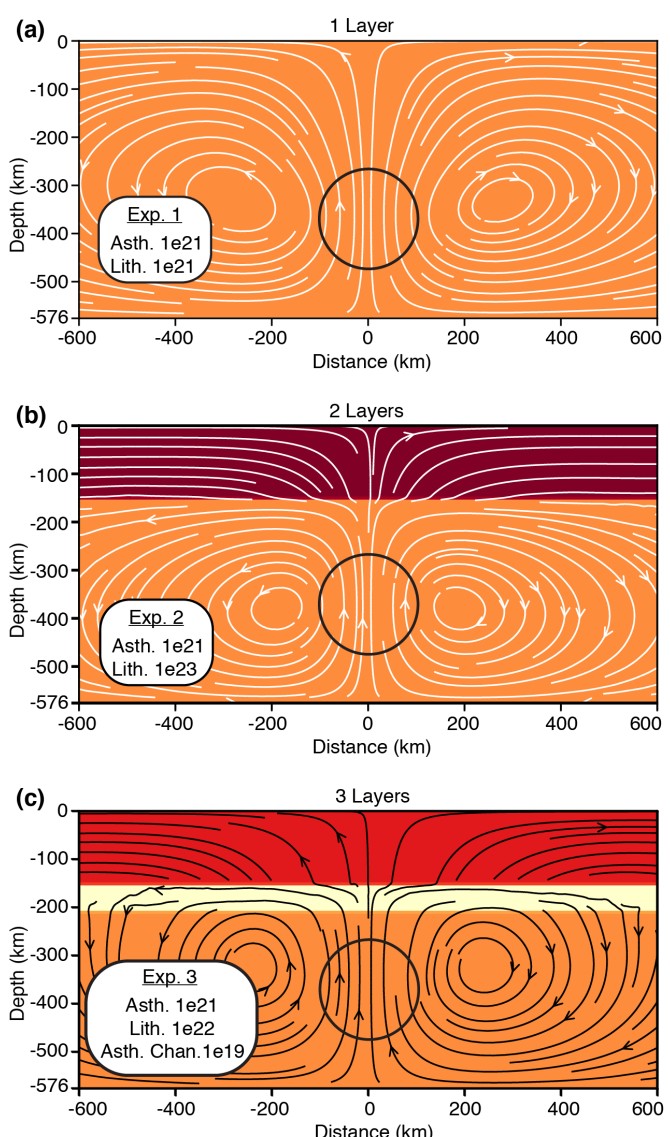


**Figure 3. Viscosity map and streamlines in a 2D cross section (*x-z* plane) along the centre of the numerical models (*y*=0 km). All experiments include an embedded sphere with 96 km radius and centred at 372 km below the surface. The sphere (i.e. black circles) has a temperature anomaly (+324 °C) giving 1% effective density difference**





**with the background mantle. The ambient fluid has 1,2 or 3 isoviscous layers for**
**Experiments 1,2 and 3 respectively.**

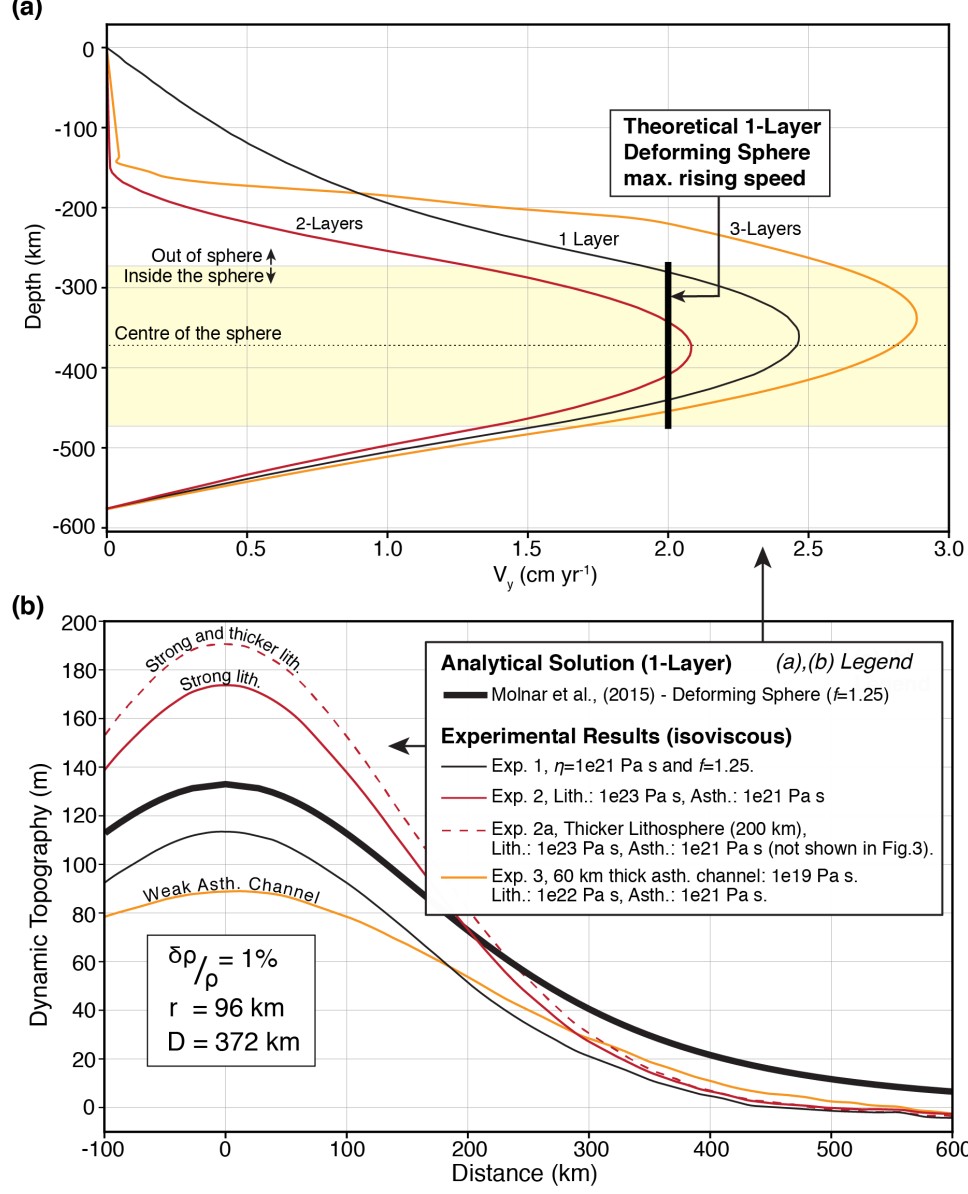


**Figure 4. (a) Vertical velocity profiles ($V_y$) along the centre, and (b) analytical solution**
**and numerical modelling results showing dynamic topography induced by a sphere of**





temperature anomaly in the mantle (*r*=96 km, $\delta\rho/\rho$ = 1%). The misfit between the
numerical model for *R*=1 and the analytical solution is due to finite space in the
numerical model compared to semi-infinite space assumed in the analytical solution
(Morgan 1965a).

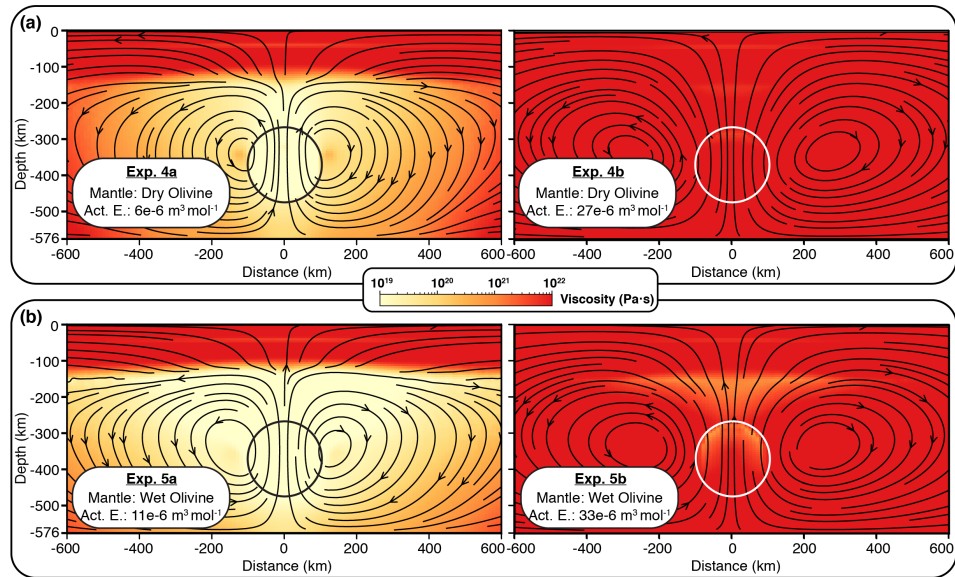


**Figure 5. Viscosity map and streamlines in a 2D cross section along the centre of the**
**numerical model (*y*=0) for Experiments using nonlinear rheologies for the crust and**
**mantle. The rising sphere is shown by black or with circles in each plot. In Experiments**
**4 and 5, the crust and mantle has visco-plastic rheologies (see Table 1 for all**
**parameters). The crust has dislocation creep of quartzite (Ranalli, 1995) rheology for**
**Experiments 4a-b and Experiments 5a-b. In the mantle, the dislocation creep of dry and**
**wet olivine rheologies are used for Experiments 4a-b and Experiments 5a-b,**
**respectively (Hirth and Kohlstedt, 2003). For each experimental set (e.g. Experiments**
**4a-b), we use lowest and highest activation volumes reported for the dry or wet olivine**
**rheology (Hirth and Kohlstedt, 2003).**



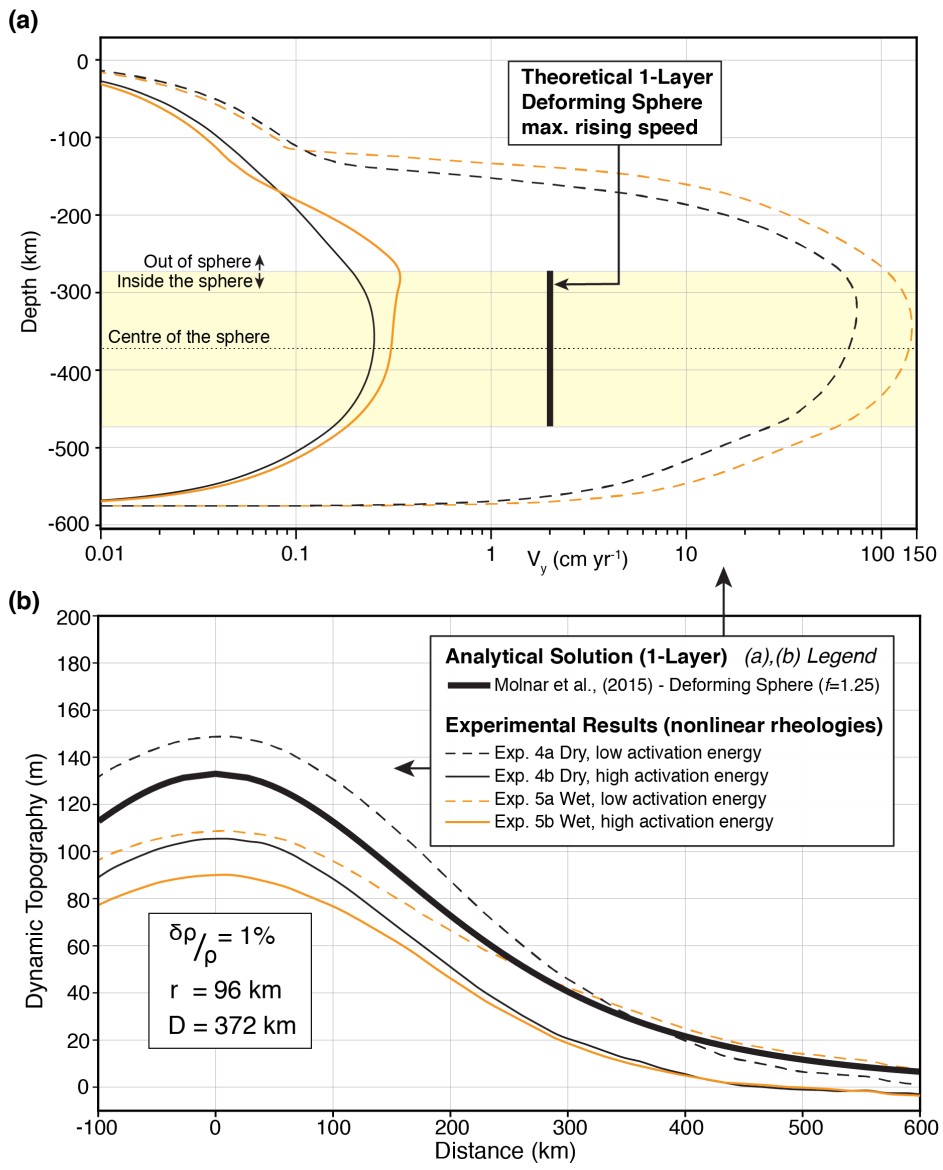


**Figure 6. (a) Vertical velocity profiles ($V_y$) along the centre and (b) dynamic topography**

**induced by a sphere of temperature anomaly ($r$=96 km, $\delta\rho/\rho$ = 1%) in the mantle that**

**has nonlinear rheology depending on temperature, pressure and strain rate.**





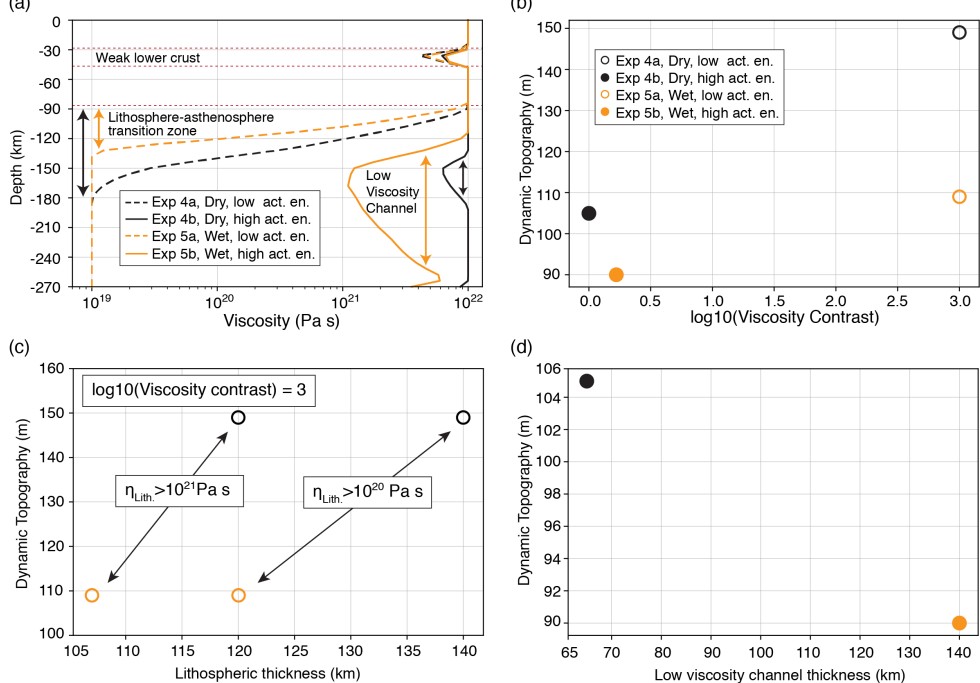

**Figure 7. Important factors affecting the amplitude of dynamic topography in Experiments 4a-b and 5a-b which have nonlinear rheologies for the crust and mantle. (a) Vertical viscosity profiles at the centre of the models. Variation in dynamic topography (b) by viscosity contrast between the lithosphere and part of the asthenosphere above the spherical temperature anomaly, (c) by lithospheric thickness at constant viscosity contrast of 3 order of magnitude assuming that lithosphere-asthenosphere boundary is at $10^{20}$ Pa s or $10^{21}$ Pa s in Experiments 4a and 5a, (d) and by thickness of low viscosity channel above the spherical anomaly and beneath the lithosphere. This low viscosity channel forms only in Experiments 4b and 5b. In Experiments 4a and 5a, the viscosity profiles show progressive variation between the lithosphere and part of the asthenosphere above the spherical anomaly.**



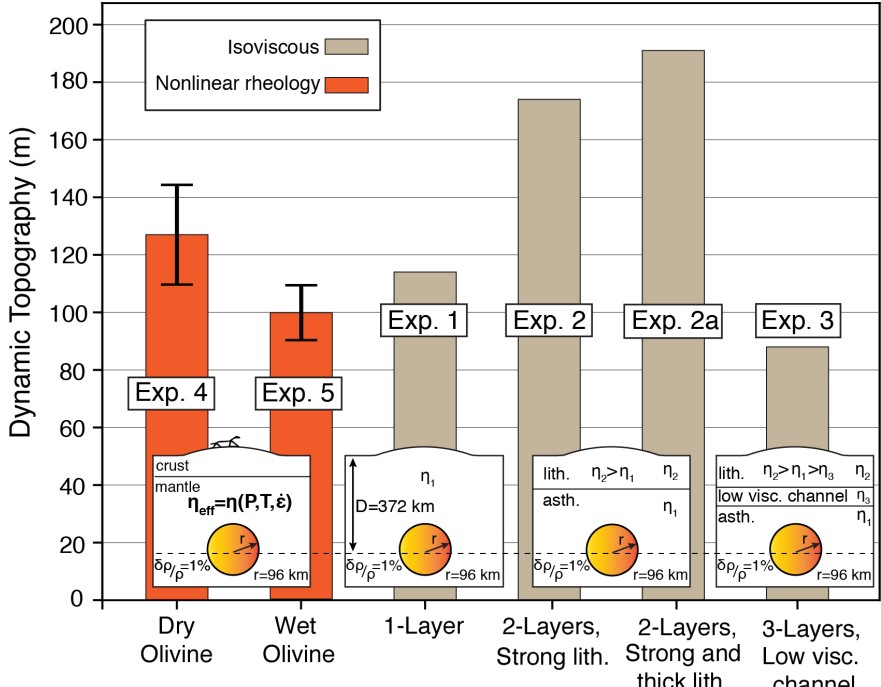

409

**Figure 8. Compilation of predicted dynamic topographies driven by a rising sphere centred at 372 km depth with 96 km radius and 1% less denser from the ambient mantle. The difference between the experiments is either due to viscosity structure (isoviscous vs. nonlinear) in the crust and mantle or thickness of the lithosphere. We also show the model configurations for each experiment. For Experiments 4 and 5, we show variation in dynamic topography from the average of experimental results for each experimental set (e.g. Experiments 4a-b) by using error bars. These variations correspond to experiments using different activation volumes reported for dislocation creep of dry and wet olivine rheologies (Hirth and Kohlstedt, 2003). In general, experiments with nonlinear rheologies having up to 3 orders of magnitude variation in viscosity in the upper mantle (between $10^{19}$ Pa s and $10^{22}$ Pa s) generally predict lesser magnitude of dynamic topography compared to experiments using isoviscous rheology. Among the experiments using nonlinear rheologies, Experiment 5 which has wet olivine**

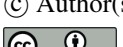



**rheology in the mantle gives lesser amplitude of dynamic topography compared to**
**Experiment 4 which has dry olivine for the same material.**


| Parameters | EXP 4a-b,5a-b Crust[1] | EXP 4a,4b Mantle[2] | EXP 5a,5b Mantle[3] |
|---|---|---|---|
| Pre-exponential factor ($MPa^{-n}s^{-1}$) | $6.7 \times 10^{-6}$ | $1.1 \times 10^{5}$ | 1600 |
| Activation energy ($kJ\,mol^{-1}$) | 156 | 530 | 520 |
| Grain size exponent | 0.0 | 0.0 | 0.0 |
| Power law exponent | 2.4 | 3.5 | 3.5 |
| Water fugacity | N.A. | N.A. | 1000 |
| Water fugacity exponent | N.A. | N.A. | 1.2 |
| Activation volume ($m^3\,mol^{-1}$) | 0.0 | $\underline{6 \times 10^{-6}}$ or $\underline{27 \times 10^{-6}}$ | $\underline{11 \times 10^{-6}}$ or $\underline{33 \times 10^{-6}}$ |
| Reference density ($kg\,m^{-3}$) | 2,700 | 3,370 | 3,370 |
| Reference temperature (K) | 293.15 | 293.15 | 293.15 |
| Initial cohesion (MPa) | 10 | 10 | 10 |
| Cohesion after weakening (MPa) | 2 | 2 | 2 |
| Initial coefficient of friction | 0.577 | 0.577 | 0.577 |
| Coefficient of friction after weakening | 0.017 | 0.017 | 0.017 |
| Saturation strain | 0.2 | 0.2 | 0.2 |
| Thermal diffusivity ($m^2\,s^{-1}$) | $1 \times 10^{-6}$ | $1 \times 10^{-6}$ | $1 \times 10^{-6}$ |
| Thermal expans. ($K^{-1}$) | $3 \times 10^{-5}$ | $3 \times 10^{-5}$ | $3 \times 10^{-5}$ |
| Compressibility ($MPa^{-1}$) | $4 \times 10^{-5}$ | 0 | 0 |
| Heat capacity ($J\,K^{-1}\,kg^{-1}$) | 1,000 | 1,000 | 1,000 |
| Radiogenic heat production ($W\,m^{-3}$) | $0.5 \times 10^{-6}$ | $0.2 \times 10^{-7}$ | $0.2 \times 10^{-7}$ |

**Table 1. Thermal and rheological parameters for all experiments. References we are**
**based on using the rheological parameters are (1) quartzite (Ranalli, 1995), (2)**
**dislocation creep of dry olivine (Hirth and Kohlstedt, 2003) and (3) dislocation creep of**
**wet olivine (Hirth and Kohlstedt, 2003). Activation volume is varied in experimental**
**sets of 4a-b, and 5a-b.**



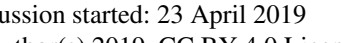
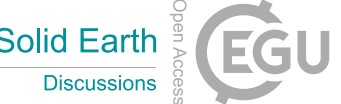

**Author contribution**


Ö.F.B designed the experiments and wrote the manuscript. P.F.R. improved the manuscript
and contributed in discussion of numerical modelling results.

**Competing interests**


The authors declare that they have no conflict of interest.

**Code and data availability**


In our experiments, we used Underworld, a free open-source code developed under the
Australian Auscope initiative.
The version of *Underworld* code we used in our study can be found at:
https://github.com/OlympusMonds/EarthByte_Underworld

To follow an open-source philosophy and promote reproducible science, our input scripts (a
suite of xml input scripts) will be available directly through the EarthByte's freely accessible
web server as well as author's GitHub repository.

**Acknowledgements**


This research was undertaken with the assistance of resources from the National
Computational Infrastructure (NCI), through the National Computational Merit Allocation
Scheme supported by the Australian Government; the Pawsey Supercomputing Centre with
funding from the Australian Government and the Government of Western Australia, and
support from the Australian Research Council through the Industrial Transformation
Research Hub grant ARC-IH130200012.





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
