# Peer review of "The Impact of Rheological Uncertainty on Dynamic Topography Predictions"

_Solid Earth, 2019_

## Referee Comment (RC1) · Bernhard Steinberger (Referee) · 27 May 2019

While I think the general idea of the paper – to study how much lateral viscosity variations (LVV) due to temperature and strain rate dependence may help to explain the discrepancy between the (higher) amplitudes of dynamic topography inferred from mantle flow and the (lower) residual topography estimates based on observations is useful in that it addresses an unresolved problem, and I also appreciate the relatively simple setup, which should help with gaining a qualitative understanding, I think the current paper suffers from several shortcomings, which limit its usefulness.

Firstly, the parts without LVV add nothing new to what is already known. Of course, I realize that these are mainly meant for comparison with the later results with LVV. But that a contrast between low-viscosity mantle and high-viscosity lithosphere leads to increased dynamic topography, and the topography gets higher the stronger and/or thicker the lithosphere is, and that an asthenospheric low-viscosity channel leads to reduced topography can all be inferred from topography kernels (see e.g. my papers from 2001 doi:10.1016/S0012-821X(01)00229-1 Fig. 2 and 2016 doi:10.1093/gji/ggw040 Fig. 3), for a broad range of depths and size of anomalies (corresponding to spherical harmonic degree). In contrast, your results are just for particular anomaly depths and (rather small) size compared to what is seen in tomography. I think for such small scales the effect of a low viscosity asthenosphere channel are stronger than for the larger scales seen by seismic tomography. E.g. in my 2016 paper Fig. 9a I find that one needs a very strong reduction in asphenosphere viscosity in order to get an appreciable reduction in topography, if anomalies are inferred from tomography. So I think the comparatively strong reductions in topography you show for a low-viscosity channel are partly misleading. Also, in my Tectonophysics paper (doi:10.1016/j.tecto.2017.11.032) I find that the largest discrepany by more than a factor 2 is at spherical harmonic degree two, whereas the discrepancy is much smaller at higher spherical harmonic degrees (i.e. smaller scales). It seems that your results could mainly explain a discrepancy at small scales, whereas the real discrepancy is at very large scales, and your results cannot explain this.

Secondly, I think the usefulness of the models with LVV is severely limited because of the limitation of viscosities to the interval 10\*\*19 Pas to 10\*\*22 Pas. In this way, model 4b is almost the same as model 1 with constant viscosity (and giving very similar amplitude), model 4a approximately corresponds to the 2-layer model with topography accordingly increased, and model 5b has a low-viscosity channel with topography accordingly reduced. What I am puzzled about, though is that case 5a gives almost the same topography as 4a although it is also two-layer (although with thinner lithosphere). I think this limitation kind of beats the purpose of introducing a realistic

rheology, because models essentially turn out to be a more complicated implementation of the easier models without LVV above. Also, I expect that without a cutoff, lowering activation enthalpy would not only lead to overall reducing viscosity, but also reducing viscosity contrasts. So, in contrast to your results I would expect a weaker lithosphere-asthenosphere contrast, and hence reduced dynamic topography for the lower activation enthalphy.

In the following are a few more consecutive comments: 1.54: as said, this large descrepancy is at the very largest scales, much larger than your model. I. 220 this equation could actually be quite simplified. Because grain-size exponent p=0 the factor d\*\*(p/n) is equal to 1 and therefore disappears. In each case, A\*\*(-1/n)\*f\_H2O\*\*(-r/n) is just a given number so you could simplify the equation in this way. I. 223 should be "volume and energy" (i.g. the other way round) I. 273 should be "wet olivine" (remove "dry"). I. 321 I don't know where I would have said in that paper that the misfit demands a scaling factor  $\sim$ 0.35, It it true that one needs to downscale shallow seismic anomalies, but I believe this has nothing to do with viscosity structure; it is rather because the thermal anomalies and corresponding seismic anomalies in the lithosphere are largely compensated by chemical anomalies, with a much smaller seismic signature. Fig 1 a: Why the results for Morgan Hard Sphere and Molnar Hard Sphere are different? I think they are both analytical results, so they should be identical. Fig. 7c: Viscosity 10\*\*20 Pas or 10\*\*21 Pas at the lithosphere-asthenosphere boundary both seems much too low to me. Table 1: For better comparison with text and eq. 5, you could also include the symbols (in those cases where you have defined them) in another column. I think the units for the pre-exponential factor should be MPa\*\*-ns\*\*-n (not -1)

minor comments: I. 28: write "from the surface" I. 65: better "dependence ... on" ?

СЗ

---

## Referee Comment (RC2) · Mark Hoggard (Referee) · 4 Oct 2019

**Review of**
**"The impact of rheological uncertainty on dynamic topography predictions: Gearing up for dynamic topography models consistent with observations"**

**October 2019**

The authors have performed numerical simulations of Stokes flow for a density anomaly in the mantle under a variety of different rheological assumptions. These simulations are benchmarked against analytical solutions for some of the simpler model setups. More complex behaviours are then explored, including using a power law rheology for which analytical solutions do not exist. The authors show that the rheological choices can have a profound impact on the observed dynamic topography observed at the surface.

The paper is mostly well written and contains a simple yet powerful illustration of some of the potential pitfalls in modelling dynamic topography. Some of the effects that are highlighted are already relatively well known, but are worth repeating and are useful in combination with the new results for the power-law rheology. My principal issue surrounds the motivation for the study, which is ostensibly concerning the amplitude mismatch between observed and predicted dynamic topography at long wavelengths (spherical harmonic degree $\sim 2$). However, I think that the model set up means that the main conclusions are probably more applicable for shorter wavelength features, and the significance for long-wavelengths mismatch remains under-explored. Nevertheless, I still think that this is an elegant illustration of some of the caveats associated with mantle convection modelling, and recommend that it be published in Solid Earth Discussions.

Mark Hoggard

Detailed comments (line numbers from the supplied .pdf):

**Main comment:**

**Discrepancy between observed and predicted dynamic topography:** As you explain in Lines 41–55 there is a mismatch between the amplitude of observed residual topography and dynamic topography predicted from simulations. Over the last few years, there has been a general focus on the long-wavelength (degree 2) components, where the driving density anomalies have comparable lateral scales to the depth of the mantle. Instantaneous flow kernals (with no lateral viscosity variations) show that the effect of features such as a low viscosity asthenosphere are less pronounced at the lower degrees than at higher degrees (shorter wavelengths). Thus, I think that the experimental set up that you are using is more suited to comparison with shorter wavelength density anomalies, and the results on long-wavelength dynamic topography predictions could turn out to be less dramatic.

Nevertheless, I think that there is also potentially an issue with amplitudes at short wavelengths. Studies that attempt to include the shallow mantle tend to predict larger dynamic topography than we observe in residual topography (e.g. Steinberger, 2016; Steinberger *et al.*, 2019; Davies *et al.*, 2019). My suspicion is that the conversion between seismic velocity and density structure is largely to blame, but your results show that the rheological assumptions may also be a significant factor. I therefore think that the motivation in your study should probably be more nuanced than it is currently written.

**Additional comments:**

**L15–17 (in abstract):** In this sentence, it is unclear that you have shown that using a power law rheology reduces dynamic topography and so potentially helps to explain this discrepancy. Please clarify, particularly the final sub-clause.

**L34:** *"…created by plate tectonic processes."* I think this should be expanded further to improve clarity. Essentially, it is dominated by isostatic topography associated with variations in the thickness and density of sediments, crust and lithospheric mantle.

**L34–39:** I think that this section is a little misleading. There are two separate types of observation: i) the absolute amplitude of dynamic topography at the present-day and ii) the rate at which it is changing. Measurements of residual topography constrain the former, as you explain in the next paragraph. The couple of sentences here on sedimentary basins are more to do with the rates of change, and in that sense are a little out of context with the rest of the manuscript. I'd suggest either clarifying this issue or removing these sentences.

**L43:** *"…isostatic components…"* is a little vague. Specifically we want to remove isostatic topography arising from sediments, crustal structure and the lithospheric mantle if we want to investigate signals arising from deeper mantle convection.

**L47:** Rather than the accuracy of the measurements, it is more whether the measurements are truly a proxy for deeper mantle contributions that depends upon the factors you highlight here.

**L59:** Repetition of *"In this paper…"*.

**L67:** Replace *"…lesser magnitude…"* with "…lower amplitudes…".

**L85:** Replace $\rho$ with $\Delta\rho$ and explain the difference between air and water-loaded dynamic topography.

**L95–96:** This is a little hard to read and would benefit from clearer grammar.

**L108:** Replace *"...normal total stress..."* with "...total normal stress...".

**L109:** *"...mass anomaly per unit length..."* – what length is this referring too?

**L111:** This needs a lead in sentence. Something like "Total normal stress can be calculated in the Fourier domain according to..."

**L122:** Start this sentence with a clause like "Although unrealistic for the Earth, under the assumption where..."

**L140:** What is the purpose of this crustal layer? Is it an elastic lid? Does it have a rheology that deforms during the simulations? Please clarify. It does not show up in the Figure pictures.

**L159–160:** Does this affect happen in all of your simulations?

**L169:** Qualify what the asthenosphere here refers to. Is it the whole of the rest of your model domain beneath the lithosphere? How is the asthenosphere defined?

**L197–199:** Good! This is a very clear and useful explanation of the cause of this behaviour.

**L225–227:** I did not know that this was generally accepted. Is this an opinion of the authors? Some back up references would be helpful. I agree that larger deviatoric stresses are thought to promote deformation by dislocation creep.

**L169:** Typo – currently reads *"...creep of wet dry olivine..."*

**Figure 1:** I think the y-axis in panel (b) would be better as dynamic topography for comparison to panel (a). Also, the key in (b) is a bit messy... A legend as in panel (a) would be clearer.

**Figure 3:** These are great, but could do with standardising to make it a truly iconic figure. Could you i) add a line above the surface showing the dynamic topography (or state the peak value), ii) make all streamlines the same colour (either white or black), iii) place the key entries in their true depth order (lith, channel, asthen). I also think it could be clearer that the relative viscosity jumps between layers are what is important, rather than absolute values, but it is fine as is.

**References**

Davies, D. R., Valentine, A. P., Kramer, S. C., Rawlinson, N., Hoggard, M. J., Eakin, C. M., & Wilson, C. R., 2019. Earth's multi-scale topographic response to global mantle flow, *Nature Geoscience*, **12**, 845–850.

Steinberger, B., 2016. Topography caused by mantle density variations: Observation-based estimates and models derived from tomography and lithosphere thickness, *Geophysical Journal International*, **205**, 604–621.

Steinberger, B., Conrad, C. P., Tutu, A. O., & Hoggard, M. J., 2019. On the amplitude of dynamic topography at spherical harmonic degree two, *Tectonophysics*, **760**, 221–228.

---

## Author Comment (AC1) · 14 Nov 2019

**Response to the Reviewer 1's Comments**

Blue: Reviewer's comment,

Black: Response to the comment,

Red: Lines added or modified in the revised manuscript

While I think the general idea of the paper – to study how much lateral viscosity variations (LVV) due to temperature and strain rate dependence may help to explain the discrepancy between the (higher) amplitudes of dynamic topography inferred from mantle flow and the (lower) residual topography estimates based on observations is useful in that it addresses an unresolved problem, and I also appreciate the relatively simple setup, which should help with gaining a qualitative understanding, I think the current paper suffers from several shortcomings, which limit its usefulness.

Firstly, the parts without LVV add nothing new to what is already known. Of course, I realize that these are mainly meant for comparison with the later results with LVV. But that a contrast between low-viscosity mantle and high-viscosity lithosphere leads to increased dynamic topography, and the topography gets higher the stronger and/or thicker the lithosphere is, and that an asthenospheric low-viscosity channel leads to reduced topography can all be inferred from topography kernels (see e.g. my papers from doi:10.1016/S0012-821X(01)00229-1 Fig. 2001 2 and 2016 doi:10.1093/gji/ggw040 Fig. 3), for a broad range of depths and size of anomalies (corresponding to spherical harmonic degree). In contrast, your results are just for particular anomaly depths and (rather small) size compared to what is seen in tomography.

We agree with the reviewer that the arguments at the beginning of the paper can be derived from radial stress or topography kernels for all wavelengths and depths (Hager and Clayton, 1989; Richards and Hager, 1989; Steinberger et al., 2001; Steinberger, 2016). As also pointed out by the reviewer, we imposed in our model a particular depth in the upper mantle and a relatively smaller anomaly size. It's true that one could follow a spherical harmonics approach in addressing dynamic topography. However, when the viscosity has lateral variations, spherical harmonic analysis becomes relatively hard to investigate analytically. When the mantle has non-linear rheology, all wavelengths of dynamic topography become coupled, and the degree of coupling should depend on how viscosity varies with temperature, pressure and strain-rate. Previous estimates using perturbation theory are insightful on understanding the impact of horizontal harmonic variations in viscosity on the dynamic topography as well the geoid (Richards and Hager, 1989), however these models assume that the lateral viscosity variations are in phase with the density anomalies, which is not necessarily the case in power-law rheology which introduces additional radial and lateral variations in viscosity as we show in our numerical models. Therefore, the variations in viscosity driven by a density anomaly in the upper mantle can reach well beyond the spatial dimensions of the embedded anomaly, and affects the mechanical lithospheric thickness. In that case, the dynamic topography varies considerably from the case where the mantle is isoviscous.

Based on above arguments, in our manuscript, we first introduced the Morgan (1965)'s analytical work on dynamic topography which uses radially layered viscosity model (up to two layers) for the Earth's interior. This helps us to easily compare it with more complex numerical models in the case for upper mantle density anomalies. Regarding the size of the anomaly, we explained our reasoning. A small radius minimizes the artefacts in the calculations and provides a better comparison with the analytical solutions carried out in an infinite half-space. The aim of this paper is not to predict the dynamic topography by using a density model derived from a seismic tomography (Steinberger et al., 2001, 2019; Flament et al., 2013), but to give insights on the first order changes in dynamic topography driven by non-linear rheology of the mantle. Having said that, we agree with the reviewer's point that the impact of larger anomalies, especially in the lower mantle, should be considered in future works.

I think for such small scales the effect of a low viscosity asthenosphere channel are stronger than for the larger scales seen by seismic tomography. E.g. in my 2016 paper Fig. 9a I find that one needs a very strong reduction in asthenosphere viscosity in order to get an appreciable reduction in topography, if anomalies are inferred from tomography. So I think the comparatively strong reductions in topography you show for a low-viscosity channel are partly misleading.

The assumptions about lithospheric thickness and radial viscosity may be effective in concluding the above mentioned argument. At the length-scales of our work (

**Figure S1:** The ratio of viscosity fields of the supplementary models having wider viscosity window of  $10^{18}$  Pa·s to  $10^{23}$  Pa·s, to the models in the manuscript using a relatively narrower viscosity window of  $10^{19}$  Pa·s to  $10^{22}$  Pa·s. The change in dynamic topography and variation in min. and max. viscosities are given in the lower-left and middle-right for each model.

---

## Author Comment (AC2) · 14 Nov 2019

**Response to the Reviewer 2's comments**

Blue: Reviewer's comment,

Black: Response to the comment,

Red: Lines added or modified in the revised manuscript

The authors have performed numerical simulations of Stokes flow for a density anomaly in the mantle under a variety of different rheological assumptions. These simulations are benchmarked against analytical solutions for some of the simpler model setups. More complex behaviours are then explored, including using a power law rheology for which analytical solutions do not exist. The authors show that the rheological choices can have a profound impact on the observed dynamic topography observed at the surface.

The paper is mostly well written and contains a simple yet powerful illustration of some of the potential pitfalls in modelling dynamic topography. Some of the effects that are highlighted are already relatively well known, but are worth repeating and are useful in combination with the new results for the power-law rheology. My principal issue surrounds the motivation for the study, which is ostensibly concerning the amplitude mismatch between observed and predicted dynamic topography at long wavelengths (spherical harmonic degree ~ 2). However, I think that the model set up means that the main conclusions are probably more applicable for shorter wavelength features, and the significance for long-wavelengths mismatch remains under-explored. Nevertheless, I still think that this is an elegant illustration of some of the caveats associated with mantle convection modelling, and recommend that it be published in Solid Earth Discussions.

Main comment:

Discrepancy between observed and predicted dynamic topography: As you explain in Lines 41–55 there is a mismatch between the amplitude of observed residual topography and dynamic topography predicted from simulations. Over the last few years, there has been a general focus on the long-wavelength (degree 2) components, where the driving density anomalies have comparable lateral scales to the depth of the mantle. Instantaneous flow kernels (with no lateral viscosity variations) show that the effect of features such as a low viscosity asthenosphere are less pronounced at the lower degrees than at higher degrees (shorter wavelengths). Thus, I think that the experimental set up that you are using is more suited to comparison with shorter wavelength density anomalies, and the results on long-wavelength dynamic topography predictions could turn out to be less dramatic.

Nevertheless, I think that there is also potentially an issue with amplitudes at short wavelengths. Studies that attempt to include the shallow mantle tend to predict larger dynamic topography than we observe in residual topography (e.g. Steinberger, 2016; Steinberger et al., 2019; Davies et al., 2019). My suspicion is that the conversion between seismic velocity and density structure is largely to blame, but your results show that the rheological assumptions may also be a significant factor. I therefore think that the motivation in your study should probably be more nuanced than it is currently written.

We are thankful to the reviewer for insightful comments and we agree with the reviewer on points made about wavelength of the dynamic topography. In the revised version of our manuscript, we put more emphasis on the fact that, shorter wavelengths of dynamic topography are being explored and long-wavelengths are currently under-explored. We also mentioned that all wavelengths become coupled in a non-

Newtonian mantle (Richards and Hager, 1989) and a more realistic rheology for the upper mantle should be considered in future works.

We realize that our paper is well-timed with a recent work by Davies and colleagues presenting that it's of critical importance to consider the non-linear viscosity structure of the lithosphere and shallow upper mantle (i.e. dependence on pressure and temperature) on global mantle convection models to accurately predict Earth's dynamic topography (Davies et al., 2019). It's worth to mention that, in our models, the viscosity also depends on strain rate, which is critical in inducing local reductions in viscosity in regions far beyond the boundaries of the embedded density anomaly (i.e. at lower part of the lithosphere). This modulates the effective mechanical thickness of the lithosphere and affects the prediction for amplitude of dynamic topography.

We also find the reviewer's comment on the conversion between seismic velocity and density interesting, and useful to mention. We briefly added a statement about it to emphasize that such uncertainty might be playing a role in predicting the amplitude of dynamic topography in global mantle convection models (Lines 53-55 in the revised manuscript). However, in the revised manuscript, we decided not to expand further on this as that would be an undertaking beyond the scope of our paper.

Additional comments:

L15–17 (in abstract): In this sentence, it is unclear that you have shown that using a power law rheology reduces dynamic topography and so potentially helps to explain this discrepancy. Please clarify, particularly the final sub-clause.

Thanks. We clarified that sentence. Now, that part of the abstract reads as "*In this paper, we use 3D numerical experiments to evaluate the extent to which the dynamic topography depends on mantle rheology. We calculate the amplitude of*

*instantaneous dynamic topography induced by the motion of a small spherical density anomaly (~100 km radius) embedded into the mantle. Our experiments show that, at relatively short wavelengths (<1,000 km), the amplitude of dynamic topography, in the case of non-Newtonian mantle rheology, is reduced by a factor of ~2 compared to isoviscous rheology.*"

L34: "…created by plate tectonic processes." I think this should be expanded further to improve clarity. Essentially, it is dominated by isostatic topography associated with variations in the thickness and density of sediments, crust and lithospheric mantle.

Thanks, we replaced this statement with the following (in Lines 36-38 in the revised manuscript): "*Because it is typically a low-amplitude and long-wavelength transient signal, it is often dwarfed by isostatic topography associated with variations in the thickness and density of sediments, crust and mantle lithosphere.*"

L34–39: I think that this section is a little misleading. There are two separate types of observation: i) the absolute amplitude of dynamic topography at the present-day and ii) the rate at which it is changing. Measurements of residual topography constrain the former, as you explain in the next paragraph. The couple of sentences here on sedimentary basins are more to do with the rates of change, and in that sense are a little out of context with the rest of the manuscript. I'd suggest either clarifying this issue or removing these sentences.

We agree with the reviewer. We removed those sentences that were out of the context with the rest of the text.

L43: "…isostatic components…" is a little vague. Specifically we want to remove isostatic topography arising from sediments, crustal structure and the lithospheric mantle if we want to investigate signals arising from deeper mantle convection.

We replaced "isostatic components" with "*isostatically compensated topography*" in Line 42 in the revised text.

L47: Rather than the accuracy of the measurements, it is more whether the measurements are truly a proxy for deeper mantle contributions that depends upon the factors you highlight here.

Thanks. We edited that sentence accordingly by removing "*the accuracy of…*". The new version is as follows (in Lines 46-48 in the revised manuscript): "*However, these residuals depend on our knowledge of the thermal and mechanical structure of the lithosphere, and therefore may not be an accurate estimation of the deeper mantle contribution to the Earth's topography.*"

L59: Repetition of "In this paper…".

Thanks, we deleted one of them.

L67: Replace "…lesser magnitude…" with "…lower amplitudes…".

Thanks, we edited that sentence.

L85: Replace $\rho$ with $\Delta\rho$ and explain the difference between air and water-loaded dynamic topography.

Now, it reads as $\Delta\rho$ rather than $\rho$ in the edited version, with a mention on the air and water-loaded case. The new version is as follows (in Line 89): "*where $\Delta\rho$ is the density difference between the mantle and air (or water assuming a sea-load when e<0)*

*(Morgan, 1965a; Houseman and Hegarty, 1987).* " We also simplified the equation a bit more (Eq. 2 in the revised text).

L95–96: This is a little hard to read and would benefit from clearer grammar.

Thanks, we simplified that sentence accordingly. We replaced "...*where C²=D²+x² and*

$f = (\eta_1 + \frac{3\eta_{sphere}}{2})/(\eta_1 + \eta_{sphere})$, *for very viscous sphere* ($\eta_{sphere} \gg \eta_1$) *f=1.5, and*

*deformable sphere* ($\eta_{sphere} \cong \eta_1$) *f<1.5.*" **with** "...*where* $C = \sqrt{D^2 + x^2}$ *and* $f = (\eta_1 +$

$\frac{3\eta_{sphere}}{2})/(\eta_1 + \eta_{sphere})$. *One can find that f=1.5 if the sphere is very viscous* ($\eta_{sphere} \gg$

$\eta_1$), *and f < 1.5 for any other case.*"

L108: Replace \\...normal total stress..." with \\...total normal stress...".

Thanks, we reordered that collection of words in that line, and in places where we use them.

L109: \\...mass anomaly per unit length..." - what length is this referring too?

Because Morgan (1965, p.6184) integrated a series of point mass sources spread continuously along a line, so that this term comes as a mass per unit length. We modified that sentence by giving more information in parenthesis (Lines 112-116 in the revised text): "*In this case, Morgan (1965a) showed (Eq. 4) that the total normal stress induced by the density anomaly is dependent on the mass anomaly per unit length ($M_{u,}$ for point sources integrated along a continuous line), the depth of the centre of the sphere (D), and marginally on the ratio of the viscosity of the convective mantle to the viscosity of the lithosphere ($R=\eta_1/\eta_2$).*"

L111: This needs a lead in sentence. Something like \Total normal stress can be calculated in the Fourier domain according to..."

We put a beginning statement to indicate that this solution is derived in Fourier domain. The following is added (Lines 116-117 in the revised text): "*The 2-layer problem is treated in Fourier domain with the resulting total normal stress as below:*"

L122: Start this sentence with a clause like \Although unrealistic for the Earth, under the assumption where..."

Thanks, we added the following at the beginning of the mentioned sentence (Line 129): "*Although an unrealistic proposition for the Earth, ...*"

L140: What is the purpose of this crustal layer? Is it an elastic lid? Does it have a rheology that deforms during the simulations? Please clarify. It does not show up in the Figure pictures.

This crustal layer exists in all models. It is visco-plastic, as the mantle, but with different viscous rheology (quartzite). The simulations were run to solve for instantaneous flow only; therefore, the defined crustal thickness (i.e. 42 km) is the same for all models. The crustal layer has been shown in Figure 2 and its physical properties were detailed in Table 1.

L159-160: Does this effect happen in all of your simulations?

We only tested the change in the sensitivity of the solution to the model geometry in a model with isoviscous rheology so that we could compare the resulting dynamic topography with the analytical solution in order to assess the boundary effects in the models. However, this effect could be slightly different for non-linear viscosities which we didn't pursue to investigate.

L169: Qualify what the asthenosphere here refers to. Is it the whole of the rest of your model domain beneath the lithosphere? How is the asthenosphere defined?

We clarified what we mean by asthenosphere, and lithosphere-asthenosphere boundary, as well as Figure 3. We prescribe a thermal gradient and the *thermal* lithosphere-asthenosphere boundary is defined by 1350 °C. We use the same thermal profile for all models, but for models using non-linear viscosity, the viscosity profile changes, so as the mechanical thickness of the lithosphere and thickness of asthenosphere. We added the following (Lines 274-278 in the revised text): "*We note that the viscosity contrast is attained by smoother transition between the lithosphere and asthenosphere (Fig. 7a, black dashed line). We infer the mechanical thickness of the lithosphere from the viscosity profiles plotted in Figure 7a, along which the lithosphere-asthenosphere transition zone shows a rapid decrease in viscosity (Conrad and Molnar, 1997).*"

L197-199: Good! This is a very clear and useful explanation of the cause of this behaviour.

Thanks. We are glad to know that our explanation of the decrease in amplitude of dynamic topography due to low-viscosity channel is useful to the readers.

L225-227: I did not know that this was generally accepted. Is this an opinion of the authors? Some back up references would be helpful. I agree that larger deviatoric stresses are thought to promote deformation by dislocation creep.

It is indeed generally accepted that in the convective mantle, low deviatoric stresses are not conducive to the activation of dislocation creep, and therefore that diffusion creep is the dominant strain mechanism (Karato and Wu, 1993, Turcotte and Schubert,

2002). In the vicinity of density anomaly, the deviatoric stresses are high enough for dislocation creep to dominate over diffusion creep. We supported our argument with references in Lines 233-236.

L169: Typo - currently reads \...creep of wet dry olivine..."
Thanks for picking this out. This was in Line 273, and we corrected it in the revised version of the manuscript (Line 292).

Figure 1: I think the y-axis in panel (b) would be better as dynamic topography for comparison to panel (a). Also, the key in (b) is a bit messy... A legend as in panel (a) would be clearer.
We modified Figure 1 and its caption based on the reviewer's suggestions.

Figure 3: These are great, but could do with standardising to make it a truly iconic figure. Could you i) add a line above the surface showing the dynamic topography (or state the peak value), ii) make all streamlines the same colour (either white or black), iii) place the key entries in their true depth order (lith, channel, asthen). I also think it could be clearer that the relative viscosity jumps between layers are what is important, rather than absolute values, but it is fine as is.

Thanks very much for the suggestions. We modified Figure 3 accordingly. The old and new versions are given below on the left and right columns, respectively.

[Figure]

**References**

Davies, D. R., Valentine, A. P., Kramer, S. C., Rawlinson, N., Hoggard, M. J., Eakin, C. M. and Wilson, C. R.: Earth's multi-scale topographic response to global mantle flow, Nat. Geosci., 12, 845–850, 2019.

Houseman, G. A. and Hegarty, K. A.: Did rifting on Australia's Southern Margin result from tectonic uplift?, Tectonics, 6(4), 515–527, doi:10.1029/TC006i004p00515, 1987.

Karato, S. and Wu, P.: Rheology of the upper mantle: a synthesis., Science, 260(5109), 771–778, doi:10.1126/science.260.5109.771, 1993.

Morgan, W. J.: Gravity anomalies and convection currents: 1. A sphere and cylinder sinking beneath the surface of a viscous fluid, J. Geophys. Res., 70(24), 6175–6187,

1965.

Richards, M. A. and Hager, B. H.: Effects of lateral viscosity variations on long-wavelength geoid anomalies and topography, J. Geophys. Res. Solid Earth, 94(B8), 10299–10313, 1989.